# A Panel of Diverse Inflammatory Biomarkers Is Not Associated with BMI-Calibrated Obesity nor with Dyslipidemia or Dysglycemia in Clinically Healthy Adults Aged 20 to 40 Years

**DOI:** 10.3390/ijerph22020207

**Published:** 2025-01-31

**Authors:** Mai S. Sater, Zainab H. A. Malalla, Muhalab E. Ali, Hayder A. Giha

**Affiliations:** 1Department of Medical Biochemistry, College of Medicine and Health Sciences (CMHS), Arabian Gulf University (AGU), Manama P.O. Box 26671, Bahrain; zainabhm@agu.edu.bh (Z.H.A.M.); muhalabae@agu.edu.bh (M.E.A.); 2Medical Biochemistry and Molecular Biology, Khartoum, Sudan; gehaha2002@yahoo.com

**Keywords:** obesity, dyslipidemia, dysglycemia, inflammation, BMI, inflammatory biomarkers

## Abstract

Objectives: Low-grade metabolic inflammation is associated with several chronic metabolic disorders, including obesity. However, no concrete evidence that supports obesity as a direct cause of chronic inflammation. This study aims to identify the association of inflammation with obesity in apparently healthy adults. Methods: In this study, 162 seemingly healthy volunteers, aged between 20 and 40 years, of comparable sex ratio, were recruited and categorized based on their body mass index (BMI) into four obesity scales: normal (N), overweight (OW), obese (OB), and severely obese (SOB). After clinical examination, fasting blood samples were collected from the study subjects for glycemic (fasting blood glucose—FBG, and HbA1c) and lipid (total cholesterol, LDL-C, HDL-C, and triacyl glycerides -TAG) profile analysis. In addition, plasma levels of a panel of diverse inflammatory biomarkers, IL6, IL8, procalcitonin (PCT), TREM1, and uPAR were analyzed by sandwich ELISA. Results: The results showed that LDLC, TAG, FBG, and HbA1c were significantly higher in the obese (OB and SOB) group, compared to the non-obese (N and OW) group, while HDLc was significantly lower. The biomarker levels were not correlated with age or significantly differed between males and females. Importantly, levels of all assessed inflammatory biomarkers were comparable between the obesity classes. Moreover, the assessed biomarkers in subjects with dyslipidemia or dysglycemia were comparable to those with normal profiles. Finally, the biomarker levels were not correlated with the obesity, glycemic, or lipidemic parameters. Conclusions: After correction for age and co-morbidities, our results deny the association of discrete obesity, probably dyslipidemia, and dysglycemia with systemic chronic inflammation. Further studies of local and systemic inflammation in non-elderly, healthy obese subjects are needed.

## 1. Introduction

Obesity is a metabolic disorder that is increasing in prevalence worldwide. It is known to increase the risk for various diseases including metabolic inflammation, insulin resistance, and cardiovascular diseases [1]. Inflammation is a normal physiological response to various noxious stimuli; however, uncontrolled and unceasing inflammation which occurs as a result of an acute inflammatory event or occurs continuously at a lower grade can have serious pathological consequences [2]. Low-grade chronic inflammation is recognized in most metabolic disorders including obesity [3,4], type 2 diabetes (T2D), metabolic syndrome, and their related complications and co-morbidities [5], and hence, this process is also described as metabolic inflammation [6]. The association between obesity and chronic inflammation was previously reported in several studies [3,7].

The main source of pro-inflammatory cytokines in obesity is adipose tissue macrophages, [3], specifically the visceral adipose tissue, which are known to induce the production of acute-phase reactants like C-reactive protein (CRP) in hepatocytes and endothelial cells [5]. Some studies have shown that abdominal/visceral adiposity is associated with elevated CRP levels, independent of body mass index (BMI), the measure of adiposity [1,8]. Similarly, another case–control study showed that CRP levels were significantly higher in individuals with abdominal adiposity than in control subjects with a comparable BMI [8].

Regarding the role of inflammation in the pathophysiology of obesity, studies showed increased activity of inflammatory pathways like c-Jun N-terminal kinase which induce the production of proinflammatory cytokines in obese subjects [9]. Moreover, inflammasomes, which are large cytosolic multiprotein complexes that initiate inflammatory response by activation of caspase-1, and secretion of potent proinflammatory cytokines, are additional inflammatory contributors to obesity [9]. The main cytokines responsible for chronic inflammation are tumor necrosis factor-*α* (TNF*α*), interleukin-6 (IL-6), and the inflammasome-activated IL-1*β*; however, IL-6, unlike TNF*α* and IL-1β, was found to be a more convenient marker of peripheral inflammation in adults with various morbidities [10]. In addition to its roles in acute phase reactions, inflammation, hematopoiesis, bone metabolism, and cancer progression, IL-6 also regulates energy homeostasis by suppressing lipoprotein lipase activity and controlling appetite and energy intake at the hypothalamic level [11]. Furthermore, IL-6 is central in the transition from acute inflammation to a chronic inflammatory state of disease, and it contributes to chronic inflammation in conditions such as obesity and insulin resistance [12]. Several other molecules are also used as inflammatory biomarkers that are associated with obesity, including the chemokine interleukin-8 (IL-8) [13], and procalcitonin (PCT) [14], the peptide precursor of the calcitonin hormone, which are used traditionally as inflammatory biomarkers in bacterial infections including sepsis [15]. However, the less reported obesity biomarkers include the Triggering Receptors Expressed on Myeloid cells-1 (TREM-1) [16], which is an immunoglobulin (Ig) superfamily transmembrane protein in humans [17]. Another inflammatory biomarker that is meagerly examined in obesity is the Urokinase-type Plasminogen Activator Receptor (uPAR) [18], also known as CD87, which is a cell surface receptor for urokinase plasminogen activator (uPA), that belongs to the lymphatic antigen-6 superfamily [19]. The biomarkers mentioned above were previously examined in T2D [20], and sepsis and septic shock [21] in our study setting.

Obesity as a cause of local inflammation is believed to have a role in systemic inflammation; one of the theories states that excessive nutrient inflow leads to enlarged, lipid-loaded adipocytes, and triggers the release of cytokines and leptin which further recruits local macrophages to adipose tissue [22]. To maintain or restore energy homeostasis, released cytokines, like IL-6, induce insulin resistance [23], as a regulatory mechanism to stop the hypertrophied adipocyte from storing lipids. The locally infiltered macrophage in response to adipocyte-derived chemokines may respond to the need for clearing the adipose tissue of dysfunctional and necrotic adipocytes. In the case of obesity as a sustained trigger, the inflammatory response does not attain its goal nor is it resolved; therefore, it turned from a local reaction to a systemic chronic state [22].

In contrast, obesity is frequently reported to be associated with mild chronic inflammation [3,4], and largely with co-morbidities that are principal causes of chronic inflammation, making the contribution of obesity to the reported inflammation dubious or bidirectional. Large data were published exploring the chronic inflammation in different clinical conditions associated with obesity, mainly T2D, metabolic syndrome, and cardiovascular disorders [5,6]. At the same time, only limited information is available about chronic inflammation in apparently healthy obese subjects [24]. Even the few articles that claimed this association were limited to metabolically healthy obesity (MHO) without consideration for the non-metabolic disorders and age, which are potential causes of inflammation [25]. In this study, we aim to confirm or reject the association of inflammation with obesity as a discrete disorder and to identify suitable biomarkers of inflammation in obese individuals. The best approach to achieve this goal is to study inflammation in obesity per se, by excluding the more frequently associated factors that lead to inflammation, mostly, co-morbidities and aging (by limiting the age between 20–40 years), which we did in this study. Since the appropriate inflammatory biomarkers in obesity are not well defined, we selected and examined an array of diverse biomarkers (IL6, IL8, PCT, TREM-1, and uPAR), of different cellular sources and pathophysiological roles, as seen above, to cover diverse etiology-based types of inflammations.

## 2. Materials and Methods

Study area and subjects: The current study was conducted in the Bahrain Defense Force (BDF) Hospital. The study subjects were volunteers, Bahraini Arabs aged 20–40 years, recruited during their routine checkups from the hospital chemistry laboratory and blood donation from the blood bank and other volunteers. The selection criteria included Arabs of Bahrani nationality, apparently healthy subjects who are not known to have any chronic or acute disorder, or receiving any regular drug treatment, except essential hypertension, which is highly prevalent in the region and was evenly prevalent in all study groups. Pregnant and lactating women were excluded from the study. Blood sample collection was planned around routine blood drawing for routine workups. A total of 162 subjects were selected for this study, and were categorized into 4 BMI classes of obesity as follows: 43 subjects with normal (N) BMI (18.5–24.9 kg/m^2^), 41 overweight (OW) subjects (25–29.9 kg/m^2^), 39 obese (OB) subjects (30–34.9 kg/m^2^), and 39 severely obese (SOB) subjects (≥35 kg/m^2^) (Table 1). Overall, less than half of the study subjects (n = 78) were obese (BMI ≥ 30). Men (n = 76) and women (n = 86) were comparable in age and BMI (Table 2). The BMI classification followed the standard WHO classification (https://www.who.int/europe/news-room/fact-sheets/item/a-healthy-lifestyle---who-recommendations (accessed on 9 January 2025)) with a minor adaption by merging obesity classes II and III into severe obesity.

Clinical data collection: For all study subjects, medical history, including history of any acute or chronic disorder and drug history, was obtained, followed by physical examination, including body weight, height, and blood pressure (BP) measurements. The body mass index (BMI) using an Excel sheet was then calculated (BMI = weight [kg] ÷ height^2^ [meters]). All data were collected on paper and later entered into SPSS/Excel database files.

Blood sampling: After 10–12 h of overnight fasting, blood was obtained from the donors by vein puncture into an EDTA tube and centrifuged. Plasma was collected into cryo-tubes and stored at –20 °C until use.

### 2.1. Clinical Chemistry Investigations

The glycemic (HbA1c and fasting blood glucose—FBG) and lipid (total cholesterol—CHL, low-density lipoprotein cholesterol—LDLc, high-density lipoprotein cholesterol—HDLc, and triacyl-glycerides—TAG) profile parameters were measured in the clinical chemistry laboratory in the hospital using automated chemo-analyzers.

For lipid profile, abnormal LDLc was considered to be ≥4.14 mmol/L (160 mg/dL), abnormal HDLc was considered to be <1 mmol/L (40 mg/dL), while abnormal TAG was ≥2.26 mmol/L (200 mg/dL). For the glycemic profile, abnormal FBG was >5.6 mmol/L (100 mg/dL), and abnormal HbA1c was ≥42 mmol/mol. The abnormal value did not include the borderline value. For estimation of the abnormal levels of test parameters, we used the reference values of the local Bahrain Defense Force (BDF) Hospital, which are consistent with the major health resources, e.g., WHO, Medscape, and the International Federation of Clinical Chemistry (IFCC). Abnormal parameters have values above the borderline ones, except for HDLc, which is below.

### 2.2. Enzyme-Linked Immunosorbent Assay (ELISA)

The levels of cytokines/chemokines and other inflammatory markers were assayed by solid-phase sandwich ELISA, using Invitrogen ELISA kits; EH2IL6 (for IL-6), KHC0081 (for IL-8), EHTREM1 (for TREM-1), and EHPLAUR (for uPAR), following the protocols provided with the kits, as previously described [20]. Similarly, presepsin was estimated using a Human Presepsin ELISA kit from MyBioSource, Catalog number MBS766136. Thermo Multiscan Spectrum Plate Reader coupled with SkanIt RE for MSS 2.4.2 software were used for measuring the plates’ absorbances.

Statistical analysis: Sigma Stat software (Systat Software Inc., version 3.5. Copyright 2006) was used for analysis. Differences between study groups were analyzed by T-test/Mann–Whitney Rank Sum Test (MW), and ANOVA/One-Way Analysis of Variance/Kruskal–Wallis One-Way Analysis of Variance on Ranks (KW). To isolate the group or groups that differ from the others, a multiple comparison procedure, All Pairwise Multiple Comparison Procedures (Dunn’s Method), was used. The correlations between variables were analyzed by Pearson Product Moment Correlation. The statistical significance was set at *p* < 0.05.

## 3. Results

### 3.1. Study Subjects’ Profile

A total of 162 subjects were enrolled in this study, 76 were males and 86 were females. The subjects were then categorized into normal (n = 43), overweight (n = 41), obese (n = 39), and severely obese (n = 39) subjects based on the BMI grading scale (18.5–24.9, 25–29.9, 30–34.9, and ≥35 kg/m^2^, respectively), and also categorized in obese and non-obese subjects. For simplification and due to great similarities in profile, the normal and overweight subjects are regrouped and described as a non-obese group, while the obese and severely obese subjects are the obese group. As seen in Table 1, the four categories were comparable in age (*p* 0916, KW), with a similar sex ratio with slight female dominance in the normal and overweight category, and male dominance in the obese category. Although the total cholesterol (CHL) was higher in the obese compared to the non-obese group, the difference was not significant (*p* 0.072); however, the LDL-c was significantly higher in the obese group and TAG was significantly and markedly higher in the obese group, while the HDL-c was significantly higher in the non-obese group. Similarly, the FBG and HbA1c were significantly higher in the severely obese category compared to the other categories.

### 3.2. Sex and Age Differences in Levels of the Inflammatory Biomarkers

As seen in Table 1 and Table 2, although males and females were comparable in number, age, and BMI, and four of the tested biomarkers, IL-8, PCT, TREM-1, and uPAR, the IL-6 was significantly higher in females (median, 25–75%) 5.92, −0.93–12.44 vs. 10.52, 3.51–15.16, *p* 0.010, MW.

### 3.3. Correlations of Demographic and Glycemic and Lipid Profiles with Biomarker Levels

None of the examined biomarkers correlated with age, BMI, and weight or FBG, HbA1c, total cholesterol, LDLc, HDLc, or TAG levels, for all correlations; the *p*-values were >0.05 (Table 3).

### 3.4. Comparison of the Inflammatory Biomarker Levels Between the Different Classes of Obesity

The clinically healthy subjects were grouped into four classes of obesity based on BMI, N (n = 43), OW (n = 41), OB (n = 39), and SOB (n = 39). The plasma levels of each of the inflammatory markers were comparable between the groups, with *p*-values of 0.199, 0.539, 0.768, 0.160, and 0.792, KW (Figure 1). Even when we re-distributed the subjects into two groups, obese (OB and SOB) and non-obese (N and OW), the five biomarkers remained comparable between them, with *p*-values of 0.745, 0.833, 0.433, 0.058, and 0.675, MW (Table 2). The differences in the levels of these biomarkers remained comparable between the obese and non-obese groups when each sex was considered separately (Table 2).

### 3.5. Comparison of Plasma Inflammatory Marker Levels Between Clinically Healthy Subjects with Normal Versus Abnormal Glycemic or Lipidemic Profiles

Subjects with abnormal HbA1c (≥42 mmol/mol, n = 17), had comparable plasma levels of IL-6, IL-8, PCT, TREM-1, and uPAR, to subjects with normal HbA1c levels, *p* 0.293, 0.652, 0.741, 0.366, and 0.767, respectively. Similarly, subjects with abnormal fasting blood glucose (FBG>5.6 mmol/L, n = 51) had comparable plasma levels of the same biomarkers as subjects with normal FBG, *p* 0.410, 0.117, 0.361, 0.290, and 0.275, MW (Figure 2). Regarding the abnormal lipid profile, LDLc (≥4.14 mmol/L), HDLc (<1 mmol/L), and TAG (≥2.26 mmol/L), also no significant differences in the levels of the tested biomarkers between those with abnormal profiles and the others with normal profiles (Figure 2). For LDLc (n =19), the *p*-values were 0.481, 0.683, 0.468, 0.948, and 0.815, for HDLc (n = 41), the *p*-values were 0.417, 0.858, 0.282, 0.146, and 0.873, and for TAG (n = 18), the *p*-values were 0.138, 0.733, 0.877, 0.565, and 0.987, MW.

## 4. Discussion

Obesity is a chronic metabolic disorder that is believed to be associated with low-grade metabolic inflammation [3,6,26]. In this study, we demonstrated that discrete obesity is not associated with inflammation based on testing a panel of five diverse inflammatory biomarkers, including IL6, IL8, PCT, TREM-1, and uPAR, which were previously reported to be associated with low-grade chronic inflammation in T2D [20]. This finding is in disagreement with several other reports that linked inflammation with obesity [7,27]. However, one unique aspect of this study is that we compared healthy subjects in the full spectrum of obesity while excluding subjects with other co-morbidities or extremes of age, which are two major confounding factors, to allow for a more accurate assessment of the link between obesity and inflammation. Therefore, inflammation related to any disease, e.g., T2D or aging, was excluded by default, as well as, the influence of drugs on the inflammatory markers’ levels was excluded since all subjects were not receiving any treatments. Moreover, the levels of the tested inflammatory biomarkers were comparable between the males and females and did not vary with age in this setting, as reported before [28].

It is well known that IL6 interferes with glucose uptake in insulin-dependent tissues, by inducing insulin resistance; therefore, it also interferes with lipolysis and oxidation of glucose and fatty acids [29,30], with a resultant loading of the adipocytes with nutrients. Insulin resistance is believed to be mediated via inflammatory and noninflammatory actions [31], which may play a role in the manifestations of obesity, and impairment of the immune control in obese adipose tissue [32]. Therefore, IL-6 leads to several metabolic disorders including metabolic syndrome and T2D [23]. Also, IL-6 plays a central role in the transition from local inflammation (localized to adipose tissue) to systemic inflammation in obesity, which is marked by raised inflammatory markers in the blood [22]. In this study, IL-6 levels were not elevated in obese subjects, or correlated with BMI or age, but it was influenced by sex (Table 2). The effect of obesity on IL-6 level was previously shown to vary with age and sex [28]; in this study, the effect of both variants was corrected for by the study design. The study subjects were selected to be in a narrow age range between 20 to 40 years, to minimize the effects of aging and its associated morbidities in inflammation. That could explain the dissociation between obesity and inflammation in this study, which was well designed to minimize the major confounders of inflammation, co-morbidities, and age. Interestingly, the only significant association in this study was the association of IL-6 with sex, as women had double the levels of this cytokine compared to men.

For the other examined inflammatory biomarkers, the plasma IL-8, PCT, TREM-1, and uPAR concentrations were not correlated with BMI or body weight nor associated with obesity, as the levels of these biomarkers were comparable between obese and non-obese groups, and in all four classes of obesity (Figure 1). This contradicts previous reports about the association of these inflammatory biomarkers with obesity. For example, IL-8 plasma concentration was previously shown to be associated with obesity in subjects with normal glucose tolerance and it was found to be correlated with the BMI, waist/hip ratio (WHR), and body fat [13]. Similarly, raised PCT, as an inflammatory marker, was found to be correlated with obesity [14]. The PCT is likely to be released from inflamed adipose tissue, reflecting the degree of endothelium-dependent vasodilatation impairment in obese individuals [14]. However, in the former study [13], the sample size was too small (a total of 75 lean and obese subjects), and there was no consideration for age which we believe is a major determinant for inflammation, while the latter study [14] was conducted in mice. In addition, increased serum PCT was recognized in obese women with polycystic ovary syndrome [33]. However, to our knowledge, there was no genuine study that linked IL-8 or PCT with isolated obesity. Also, only limited studies linked TREM-1 or uPAR with obesity. In one study, TREM-1 was shown to be raised significantly in blood and tissue biopsies from obese subjects, suggesting a possible role in obesity pathophysiology [34]. However, the previous study was not designed to exclude co-morbidities or age as confounders. Finally, an association of uPAR levels with abdominal obesity [35] and WHR but not BMI, was previously reported [36]. The dissociation of obesity from inflammation in this study, in contrast to other studies, is more likely due to the exclusion of the inflammation confounders, the obesity co-morbidities, and aging. The impact of null findings in this study, if confirmed by other similar studies, might add to our understanding of the pathophysiology of obesity and therefore its management and prevention of the progression to unhealthy obesity, probably by exercise, diet, mindfulness, and meditation that reduce both obesity and inflammation.

The other finding of this study was the lack of association of inflammation with dyslipidemia and dysglycemia in clinically healthy asymptomatic subjects. It is worth noting that both abnormalities are usually associated with obesity. The association of dyslipidemia with obesity was previously reviewed with an explanation of the mechanism of this association [37]. In this study, the LDLc and TAG were significantly higher in obese subjects (OB and SOB) compared to non-obese subjects (N and OW), while HDLc was significantly higher in the non-obese group as seen in Table 1. The bidirectional association of dyslipidemia with inflammation was reported before [38]; however, in the present study, the plasma concentrations of the examined inflammatory biomarkers did not differ between the clinically healthy subjects with normal vs. abnormal lipidemic profiles, as seen in Figure 2. Subjects with abnormally high LDLc, or TAG and subjects with abnormally low HDLc, had comparable plasma concentrations of IL-6, IL-8, PCT, TREM-1, and uPAR, with subjects with normal levels of the three lipidemic parameters, regardless of the BMI (Figure 2C–E). That was not unexpected in this study since the same inflammatory biomarkers were comparable between all the obesity classes mentioned above. This finding contrasts a previous study that demonstrated a significant link between inflammation and dyslipidemia [39].

The association and contribution of obesity to dysglycemia, namely hyperglycemia, was discussed before [40]. In the current study, subjects with abnormally raised HbA1c or FBG had comparable concentrations of the above-described panel of inflammatory biomarkers compared to subjects with normal levels of both parameters (Figure 2A,B). Similar to the bidirectional relationship between dyslipidemia and inflammation, studies have demonstrated a bidirectional association between hyperglycemia and inflammation [41,42]. However, to the best of our knowledge, the current study is the first to examine the association of inflammation with isolated hyperglycemia, independent of diabetes mellitus.

For the interplay between obesity, dyslipidemia, and dysglycemia with low-grade chronic inflammation, several confounding factors need to be considered, namely co-morbidities, aging, and drug therapies, which might be the actual stimulants to the immune system and liberation of the inflammatory markers. Current evidence strongly suggests that T2D marked by hyperglycemia is an inflammatory disease in which inflammation is mediated by obesity-linked insulin resistance and hyperlipidemia [43]. In the present study, the obese subjects had significantly different lipidemic and glycemic profiles compared to the non-obese subjects, which supports the link between obesity, dyslipidemia, and dysglycemia, but what is unique is that this link exists in the absence of diabetes mellitus, as the study subjects are clinically healthy obese subjects.

Finally, the dissociation of inflammation from obesity, dyslipidemia, or dysglycemia suggests that the association between the former disorders is conditioned with the development of frank diabetes mellitus and clinical symptoms. Alternatively, it may be that in clinically healthy obese subjects, the inflammation is local and is limited to the adipose and other tissues but does not reach the systemic level; therefore, local adipose tissue testing for inflammation can explain some of our findings. Another explanation for this dissociation could be the sample size which might not be large enough to reveal trivial inflammation. The latter might be the least possible explanation but remained as one of the limitations of this study; therefore, we suggest running more studies with larger sample sizes and examining a broader and more diverse panel of inflammatory markers, by using an ultra-sensitive technique to measure the biomarkers. Moreover, a longitudinal cohort study including data about dietary habits, physical activity, and stress control might disclose the transition from local, low, or no inflammation to systemic inflammation observed in other studies.

In conclusion, obesity per se is unlikely to induce a measurable systemic inflammatory response; similarly, discrete dyslipidemia and dysglycemia are unlikely to induce chronic inflammation in asymptomatic healthy subjects. This study showed that non-elderly adults with obesity who are clinically healthy are less likely to have chronic inflammation in contrast to studies that linked obesity with chronic inflammation. Probably most of the other studies were not restrictive in the selection of the study subjects for testing obesity as a discrete disorder, since obesity is largely associated with aging and several co-morbidities, e.g., T2D, metabolic syndrome, cardiovascular disorders, autoimmune diseases, cancer, degenerative diseases. However, more studies in different settings and using more inflammatory markers are needed.

## Figures and Tables

**Figure 1 ijerph-22-00207-f001:**
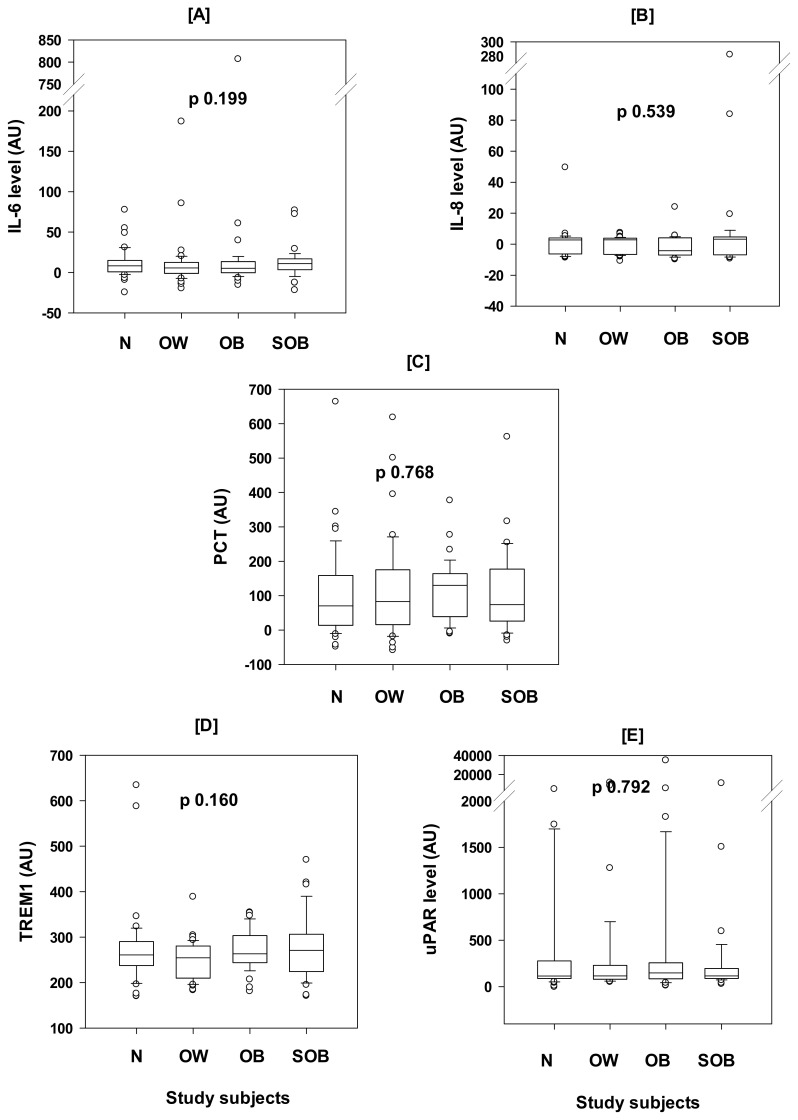
Comparison of the median (25–75%) plasma concentrations (AU = arbitrary units) of five inflammatory biomarkers, IL-6 [**A**], IL-8 [**B**], procalcitonin (PCT) [**C**], TREM-1 [**D**], and uPAR [**E**], between four BMI-calibrated obesity classes (normal (N), overweight (OW), obese (OB), and severely obese (SOB)), of healthy non-elderly adults (20–40 years). There were no significant differences in the levels of the tested inflammatory biomarkers between the different obesity classes, using the Kruskal–Wallis One-Way Analysis of Variance on Ranks (KW), *p*-values are shown in the figure. The horizontal line within each bar is the median value, the bottom and top lines of the bar are the 25% and 75%, respectively, caps of the lower and upper vertical lines are the 5% and 95% percentiles, and the open circles are outliers. The biomarker concentrations are shown in Table 4 as a figure extension.

**Figure 2 ijerph-22-00207-f002:**
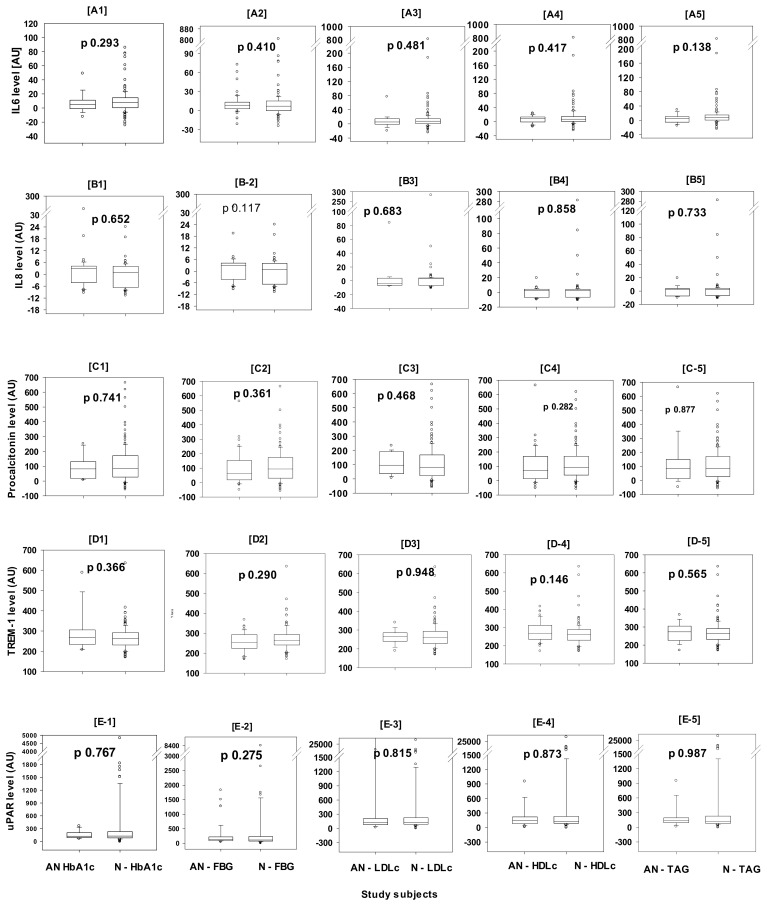
The figure shows the inflammatory response to dyslipidemia and dysglycemia in healthy non-elderly subjects (20–40 years), in the full spectrum of BMI-calibrated obesity. Comparison of the median (25–75%) plasma concentrations of five inflammatory biomarkers, IL-6 [**A1**–**A5**], IL-8 [**B1**–**B5**], procalcitonin (PCT) [**C1**–**C5**], TREM-1 [**D1**–**D5**], and uPAR [**E1**–**E5**], using arbitrary unit (AU), between subjects with normal vs. abnormal HbA1c [**A1**–**E1**], fasting blood glucose (FBG) [**A2**–**E2**], low-density lipoprotein cholesterol (LDLc) [**A3**–**E3**], triacyl-glycerides (TAG) [**A4**–**E4**], and high-density lipoprotein cholesterol (HDLc) [**A5**–**E5**]. There were no significant differences in the concentrations of any of the five tested inflammatory biomarkers between the subjects with normal vs. abnormal glycemic or lipidemic profiles, using Mann–Whitney Rank Sum Test (MW), *p*-values are shown in the figure, which were all >0.05. Within the bars, the horizontal line is the median value, the bars’ top and bottom lines are 25% and 75%, respectively, caps of the lower and upper vertical lines are 5% and 95%, and the open circles are outliers. Note: The abnormal parameters were determined based on the local reference values, which were almost consistent with international values.

**Table 1 ijerph-22-00207-t001:** The study subjects’ obesity classes, numbers, demographic characteristics, and, lipid, and glycemic profiles.

Parameters	Overall	Normal 18.5 to 24.9	Overweight 25 to 29.9	Obese 30 to 34.9	Severely Obese ≥ 35 (kg/m^2^)	*p*-Value KW
Number	162	43	41	39	39	
Age (years)	29.99 ± 5.65	29.977 ± 5.97	30.17 ± 5.81	30.31 ± 5.38	29.51 ± 5.59	0.916 *
Sex (M/F)	76/86	18/25	17/24	22/17	19/20	
BMI (kg/m^2^)	30.46 ± 7.921	22.2, 20.3–23.7	26.6, 25.55–27.70	32.8, 31.5–33.4	41.4, 37.2–45.3	<0.001
Weight (kg)	84.61 ± 23.65	60.0, 55.0–65.0	70.0, 66.5–80.0	91.0, 84.0–101.0	110.0, 101.0–129.0	<0.001
Lipid profile (mmol/L)						
TC (mmol/L)	4.94 ± 1.04	4.5, 4.2–5.10	4.9, 4.415–5.4	5.2, 4.2–5.77	5.0, 4.4–6.0	0.073
LDL-c (mmol/L)	3.02 ± 0.96	2.50, 2.20–3.13	2.83, 2.44–3.28	3.31, 2.39–4.00	3.30, 2.58–3.90	0.003
HDL-c (mmol/L)	1.19 ± 0.34	1.39 ± 0.340	1.304 ± 0.327	1.014 ± 0.249	1.008 ± 0.265	<0.001 *
TAG (mmol/L)	1.43 ± 1.29	0.90, 0.60–1.19	1.06, 0.81–1.40	1.33, 1.01–1.72	1.41, 0.90–1.91	<0.001
Glycemic profile						
FBG (mmol/L)	5.54 ± 1.39	5.15, 4.90–5.50	5.09, 4.80–5.45	5.40, 4.89–5.82	5.70, 5.16–6.28	0.002
HbA1c (mmol/mol)	36.75 ± 6.07	34.0, 32.0–37.0	37.0, 34.0–39.0	37.0, 34.0–39.0	39.0, 36.0–41.0	<0.001

Note: *p*-values for comparisons between the classes of obesity; N (normal), OW (overweight), OB (obese), and SOB (severe obesity). All comparisons were executed by Kruskal–Wallis One-Way Analysis of Variance on Ranks (KW), except the ones with asterisks (*) were executed by One-Way Analysis of Variance.

**Table 2 ijerph-22-00207-t002:** Comparisons of the examined inflammatory biomarker levels: A. males vs. females, B. non-obese vs. obese males and females, C. non-obese (all) vs. obese (all) subjects.

A. Biomarkers	Males (76)		Females (86)			*p*-Value
Age (y)	30.0, 26.0–35.0		29.0, 24.75–35.00			
BMI (kg/m^2^)	30.55, 25.05–35.35		27.95, 24.08–34.13			0.418
IL-6 (AU)	5.92, −0.93–12.44		10.52, 3.51–15.16			0.010
IL-8 (AU)	2.54, −6.58–3.93		2.83, −6.56–4.12			0.624
PCT (AU)	76.20, 21.23–167.53		88.68, 27.58–171.55			0.901
TREM1 (AU)	263.62, 230.31–292.82		261.62, 232.30–291.49			0.938
uPAR (AU)	117.79, 85.48–217.09		127.07, 82.86–229.07			0.575
**B.**	Males			Females		
	Non-obese	Obese	*p*-value	Non-obese	Obese	*p*-value
Number	35	41		49	37	
Age	29.0, 26.0–36.0	30.0, 26.5–34.5	0.917	29.0, 24.5–35.0	29.0, 24.5–34.5	0.854
IL-6	6.04, −1.68–13.07	5.85, −0.64–12.24	0.681	9.34, 3.54–14.78	11.18, 2.76–16.42	0.635
IL-8	2.83,−6.50–3.96	−4.16, −7.06–3.92	0.425	2.80, −6.57–3.97	3.14, −6.53–4.37	0.622
PCT	59.0, 13.1–136.2	99.6, 40.4–169.3	0.122	93.8, 48.1–176.0	82.1, 24.0–153.9	0.879
TREM1	257.85, 43.06–7.28	271.04, 48.3–7.54	0.216	256.6, 224.6–284.4	270.7, 240.6–307.1	0.128
uPAR	117.8, 96.3–261.1	120.5,77.9–213.6	0.457	112.2, 80.5–228.1	146.2, 93.1–287.0	0.201
**C.**	Non-obese (males and females)		Obese (males and females)			
No	84		78			
IL-6	6.729, 0.73–13.484		8.224, 0.181–15.015			0.745
IL-8	2.816, −6.508–3.933		2.189, −6.866–4.334			0.833
PCT	75.522, 15.278–173.168		95.119, 34.892–164.976			0.433
TREM1	256.595, 224.897–282.760		269.028, 236.245–304.566			0.058
uPAR	115.413, 81.515–229.073		131.013, 86.868–220.242			0.675

Statistical tests; Whitney Rank Sum Test (MW).

**Table 3 ijerph-22-00207-t003:** Correlations of the age, obesity (BMI/weight), glycemic, and lipidemic parameter levels as possible confounders with inflammatory biomarker levels.

Parameters	Correlation	IL-6	IL-8	Procalcitonin	TREM-1	uPAR
Age (years)	CC	−0.0704	0.144	0.0764	0.0110	−0.0176
	p	0.373	0.0683	0.334	0.889	0.828
BMI (kg/m^2^)	CC	0.00634	0.0578	−0.0175	0.119	0.00214
	p	0.936	0.465	0.825	0.131	0.979
Weight (Kg)	CC	−0.0407	0.0481	−0.0191	0.0822	−0.0374
	p	0.607	0.543	0.809	0.299	0.645
FBG (mmol/L)	CC	−0.0161	0.0408	0.0350	−0.0140	−0.0431
	p	0.839	0.606	0.658	0.860	0.596
HbA1c (mmol/mol)	CC	−0.0166	0.0122	−0.00424	−0.0188	−0.0186
	p	0.833	0.878	0.957	0.812	0.819
CHL (mmol/L)	CC	−0.0806	−0.0301	−0.0100	−0.0784	−0.0596
	p	0.308	0.704	0.899	0.321	0.463
LDL-c (mmol/L)	CC	−0.0856	−0.0278	−0.00528	0.0130	−0.0621
	p	0.2799	0.726	0.947	0.869	0.444
HDL-c (mmol/L)	CC	0.0287	−0.0509	−0.104	−0.123	0.0262
	p	0.717	0.520	0.190	0.119	0.747
TAG (mmol/L)	CC	−0.0593	0.0381	0.0814	−0.0391	−0.0575
	p	0.453	0.630	0.303	0.622	0.47

Note: the statistical test was Pearson Product Moment Correlation. CC: correlation coefficient, p: *p*-value.

**Table 4 ijerph-22-00207-t004:** An extension for Figure 1: the plasma levels (arbitrary units) of the five examined inflammatory biomarkers in the four obesity classes shown in the figure.

Markers	Normal	Overweight	Obese	Severely Obese
IL-6	8.29, 0.80–15.00	5.72, −1.02–12.56	5.27, −0.24–13.56	11.18, 3.54–17.04
IL-8	2.80, −6.31–4.09	2.83, −6.55–3.85	−4.16, −6.99–4.16	3.25, −6.84–4.70
PCT	70.26, 13.52–158.59	82.76, 15.69–175.00	129.73, 38.97–163.70	73.71, 25.69–177.37
TREM1	260.76, 237.45–290.20	254.73, 209.99–280.53	263.34, 243.83–303.45	271.13, 224.33–306.46
uPAR	114.74, 87.67–276.52	116.09, 80.31–229.57	147.32, 84.735–256.36	116.45, 87.58–195.74

## Data Availability

The datasets used and/or analyzed during the current study are available from the corresponding author upon reasonable request.

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
