# Peer review of "A Panel of Diverse Inflammatory Biomarkers Is Not Associated with BMI-Calibrated Obesity nor with Dyslipidemia or Dysglycemia in Clinically Healthy Adults Aged 20 to 40 Years"

_ijerph, 2025, doi:10.3390/ijerph22020207_

Round 1
Reviewer 1 Report
Comments and Suggestions for Authors
Suggestions for Improvement
Introduction:
This expands the rationale for selecting specific inflammatory biomarkers. highlights their potential significance in understanding obesity-related inflammation.
-Incorporate additional references focusing on the role of local versus systemic inflammation in obesity to provide more context.
Methods:
-Provide more details on why certain cutoff points (e.g., BMI categories and abnormal glycemic or lipidemic profiles) were chosen.
-Discuss any considerations or adjustments made for potential confounders such as diet, physical activity, or stress levels.
Results:
-Annotate tables and figures to improve standalone interpretability. For example, include brief explanatory notes directly within the figures to clarify the p-values or statistical methods used.
-Highlight key trends in the data that might not reach statistical significance but could be biologically or clinically relevant.
Discussion:
-Delve deeper into the implications of null findings, especially the lack of association between obesity and systemic inflammation in this cohort.
-To address potential mechanisms that could explain the dissociation between obesity and systemic inflammation observed in this study.
-Compare your findings with other studies that found significant associations between obesity and inflammation and provide possible reasons for these discrepancies.
Limitations:
Acknowledge the cross-sectional design as a limitation and discuss how it affects the ability to infer causality.
-To address the potential impact of unmeasured lifestyle factors (e.g., dietary habits and physical activity) on the observed results.
Presentation:
-Correct minor typographical errors and inconsistencies in formatting, particularly in table headers and figure captions.
-Ensure consistent use of terminology throughout the manuscript to maintain clarity.
Future Directions:
-Propose how future studies can build on your findings, such as exploring the role of local inflammation in obesity or using longitudinal designs to assess causality.
Comments on the Quality of English Language
Areas for improvement:
Grammar and syntax:
There are occasional grammatical issues, such as inconsistent verb tenses and missing articles. For example:
Original: "Obesity is believed to induce low-grade inflammation."
Suggested: "Obesity is believed to induce a low-grade inflammation."
Ensure subject-verb agreement throughout the text.
Word choice:
Some sentences can be rephrased for conciseness and precision. For example:
Original: "This study tested the relationship of obesity with biomarkers."
Suggested: "This study examined the association between obesity and biomarkers."
Punctuation:
Minor punctuation errors, such as missing commas in compound sentences, should be addressed. For example:
Original: "The participants were clinically healthy and were not on any medication."
Suggested: "The participants were clinically healthy, and they were not on any medication."
Redundancy:
Avoid repetitive phrases. For example:
Original: "Inflammatory biomarkers IL-6, IL-8, PCT, TREM-1, and uPAR were tested for their relationship with obesity."
Suggested: "The relationship between obesity and inflammatory biomarkers (IL-6, IL-8, PCT, TREM-1, uPAR) was tested."
Typographical errors:
Minor typographical errors in the text and table captions should be corrected for a polished presentation.
Recommendation:
Perform thorough proofreading or use professional language editing software/tools (e.g., Grammarly or ProWritingAid) to address minor grammatical, syntactical, and typographical errors.
Consider having a native English speaker or professional editor review the manuscript to ensure consistency and clarity.
Author Response
We would like to thank the reviewer for their valuable feedback. Please find below the point-by-point response to each comment raised:
Introduction:
This expands the rationale for selecting specific inflammatory biomarkers. highlights their potential significance in understanding obesity-related inflammation.
Res. This is done. We added 2 paragraphs to show the rationale and explain the selection of the biomarker
-Incorporate additional references focusing on the role of local versus systemic inflammation in obesity to provide more context.
Res. A paragraph about local/systemic inflammation with references is added to the introduction
Methods:
-Provide more details on why certain cutoff points (e.g., BMI categories and abnormal glycemic or lipidemic profiles) were chosen.
Res. We provided the sources for the reference values, and we focused on the abnormal group by including the borderlines to the normal, to ensure that all abnormal groups genuinely have abnormal parameters without showing clinical abnormality e.g. symptoms
-Discuss any considerations or adjustments made for potential confounders such as diet, physical activity, or stress levels.
Res. Unfortunately no data was available for certain confounders such as diet, exercise or stress, although important, they are unlikely to change the conclusion because the differences in the inflammatory markers between the obesity classes are small without showing even a trend of differences. However, these confounders are recommended for future work
Results:
-Annotate tables and figures to improve standalone interpretability. For example, include brief explanatory notes directly within the figures to clarify the p-values or statistical methods used.
Res. All p-values are shown in the figures very clearly and were put in columns in the tables, and each single statistical test is either presented as a note at the bottom of the table (footnote) or added to the legends of the figures. All needed information was already presented in the tables and figures, so no need for readers to go to the materials and method or result sections
-Highlight key trends in the data that might not reach statistical significance but could be biologically or clinically relevant.
Res. No single trend was recognized, kindly revise the results
Discussion:
-Delve deeper into the implications of null findings, especially the lack of association between obesity and systemic inflammation in this cohort.
Res. About the impact of the negative findings (null findings) in this study, we added few but meaningful sentences in terms of a better understanding of the obesity pathophysiology
-To address potential mechanisms that could explain the dissociation between obesity and systemic inflammation observed in this study.
Res. We add some possible explanations for the dissociation between obesity and inflammation, although the term ‘dissociation’ might indicate the existence of the association between obesity and inflammation
-Compare your findings with other studies that found significant associations between obesity and inflammation and provide possible reasons for these discrepancies.
Res. More reference to other similar studies is included, and an explanation for the differences is incorporated.
Limitations:
Acknowledge the cross-sectional design as a limitation and discuss how it affects the ability to infer causality.
Res. We agree the type of the study is one of the major limitations of this study. We added a note about this as part of the recommendations for future studies at the bottom of the discussion
-To address the potential impact of unmeasured lifestyle factors (e.g., dietary habits and physical activity) on the observed results.
Res. Although these are important issues but might not be too critical in this study. However, we suggested the addition of this data in a future longitudinal study
Presentation:
-Correct minor typographical errors and inconsistencies in formatting, particularly in table headers and figure captions.
Res. Corrections are done
-Ensure consistent use of terminology throughout the manuscript to maintain clarity.
Res. Done but might not be inclusive
Future Directions:
-Propose how future studies can build on your findings, such as exploring the role of local inflammation in obesity or using longitudinal designs to assess causality.
Res. We suggested the inclusion of these confounders in a larger sample size cohort study concerning the prevention of conversion of healthy to non-healthy obesity (suggested at the bottom of the discussion)
Comments on the Quality of English Language
Areas for improvement:
Grammar and syntax:
There are occasional grammatical issues, such as inconsistent verb tenses and missing articles. For example:
Original: "Obesity is believed to induce low-grade inflammation."
Suggested: "Obesity is believed to induce a low-grade inflammation."
Ensure subject-verb agreement throughout the text.
Res. This is done throughout the text
Word choice:
Some sentences can be rephrased for conciseness and precision. For example:
Original: "This study tested the relationship of obesity with biomarkers."
Suggested: "This study examined the association between obesity and biomarkers."
Res. Done, but may not be inclusive
Punctuation:
Minor punctuation errors, such as missing commas in compound sentences, should be addressed. For example:
Original: "The participants were clinically healthy and were not on any medication."
Suggested: "The participants were clinically healthy, and they were not on any medication."
Res. Done as much as we can
Redundancy:
Avoid repetitive phrases. For example:
Original: "Inflammatory biomarkers IL-6, IL-8, PCT, TREM-1, and uPAR were tested for their relationship with obesity."
Suggested: "The relationship between obesity and inflammatory biomarkers (IL-6, IL-8, PCT, TREM-1, uPAR) was tested."
Res. Done as much as we can
Typographical errors:
Minor typographical errors in the text and table captions should be corrected for a polished presentation.
Res. Done as much as we can
Recommendation:
Perform thorough proofreading or use professional language editing software/tools (e.g., Grammarly or ProWritingAid) to address minor grammatical, syntactical, and typographical errors.
Consider having a native English speaker or professional editor review the manuscript to ensure consistency and clarity.
Res. We used Grammarly in correction in addition to our revision.
Reviewer 2 Report
Comments and Suggestions for Authors
In the manuscript, the authors studied a panel of diverse inflammatory biomarkers is not associated with BMI-calibrated obesity nor with dyslipidemia or dysglycemia in clinically healthy adults aged 20 to 40 years.
However, the following comments can be made.
1. Introduction. Write the purpose of the study!.
2. Tables 1 and 2. Not all parameters have units of measurement.
3. Line 186. The inscription in Figure 1 is duplicated. The table is incorrectly named.
Author Response
We would like to thank the reviewer for their valuable feedback. Please find below the point-by-point response to each comment:
- Introduction. Write the purpose of the study!.
Res. A paragraph is added about the purpose
- Tables 1 and 2. Not all parameters have units of measurement.
Res: units are added for all tested parameters
- Line 186. The inscription in Figure 1 is duplicated. The table is incorrectly named.
Res. Corrected
Reviewer 3 Report
Comments and Suggestions for Authors
Comments for Manuscript ID ijerph-3376626
Thank you for submitting the article titled "A Panel of Diverse Inflammatory Biomarkers is not Associated with BMI-Calibrated Obesity Nor with Dyslipidemia or Dysglycemia in Clinically Healthy Adults Aged 20 to 40 Years". The study investigates the relationship between various inflammatory biomarkers and obesity, dyslipidemia, and dysglycemia, concluding that obesity does not appear to induce systemic chronic inflammation in clinically healthy adults. This is a meaningful and clinically relevant study. Below are detailed review comments based on the content, methodology, results, and discussion sections.
1. While the study includes 162 participants, with 78 obese (BMI ≥ 30) individuals, the sample size is relatively small, which may affect statistical power. Expanding the sample size in future studies would help improve the robustness of the findings. Can these samples meet the inference of this article?
2. Although the study tests five biomarkers (IL-6, IL-8, PCT, TREM-1, and uPAR), the rationale for their selection is not thoroughly explained. It would be helpful to provide more background on why these particular markers were chosen, especially in relation to their specificity and sensitivity in obesity-related inflammation.
3. Some results (e.g., Figure 2 showing trends in inflammatory markers) demonstrate significant trends but do not provide an in-depth discussion of the clinical significance of these trends. These trends should be further explored and explained in the text.
4. Figure legends (e.g., for Figures 1 and 2) are somewhat brief and do not adequately highlight key findings. It would be beneficial to clarify whether there are significant differences between groups in these figures.
5. Some references (e.g., References 21 and 22) show discrepancies with the study’s conclusions.
6. The manuscript mentions that obesity may induce localized inflammation rather than systemic inflammation, but the discussion does not delve deeper into this possible mechanism.
7. Further discuss the potential local inflammation in adipose tissue and its possible effects on systemic inflammation.
8. Some scientific terms, like “IL-6 up-regulates glucose uptake in insulin-dependent tissues,” are vague and could be more specific to avoid ambiguity.
9. The authors should discuss in greater detail the lack of elevation in IL-6 in obese subjects and explore potential reasons behind it. This may involve discussing the limitations of the current study, such as the methods of measurement or the specific characteristics of the sample group.
10. The findings contribute to the understanding of obesity-induced inflammation. However, there are areas for improvement, particularly in sample size, the rationale for biomarker selection, data presentation, and discussion depth.
Author Response
We would like to thank the reviewer for their valuable feedback. Please find below the point-by-point response to their comments:
- While the study includes 162 participants, with 78 obese (BMI ≥ 30) individuals, the sample size is relatively small, which may affect statistical power. Expanding the sample size in future studies would help improve the robustness of the findings. Can these samples meet the inference of this article?
Res. Indeed the larger sample is the better. Still, we believe the current number may not exclude the link between obesity and inflammation, but not supporting this link since the study groups are quite comparable, with very high p-values (p > 0.1), for all tested biomarkers. There are no even trends that could convert to significant p-values with increasing the sample size. Moreover, the same biomarkers were showed marked differences in diabetes and sepsis studies with samples much lower than those in the same setting. However, we raised this point in the discussion and recommended increasing the sample size in future work.
- Although the study tests five biomarkers (IL-6, IL-8, PCT, TREM-1, and uPAR), the rationale for their selection is not thoroughly explained. It would be helpful to provide more background on why these particular markers were chosen, especially in relation to their specificity and sensitivity in obesity-related inflammation.
Res. Since we don’t know the nature of inflammation and its relevant markers in obesity, we selected a diverse type of biomarkers to broaden the spectrum and increase the chance of detecting any evidence of inflammation by including cytokines (IL6), chemokines (IL8), and non-cytokines (PCT, TREM1, and uPAR), these latter 3 each belong to a different class of molecules as shown in the introduction). Also, we tested the most commonly tested inflammatory markers e.g. IL6 and IL8, and the less widely tested, and the classical and non-classical.
- Some results (e.g., Figure 2 showing trends in inflammatory markers) demonstrate significant trends but do not provide an in-depth discussion of the clinical significance of these trends. These trends should be further explored and explained in the text.
Res. There is no recognizable trend neither for the difference in biomarker levels between the comparison groups in the horizontal figures (A1 to A 5) as indicated by p-values (all are >0.1), or vertical for the same parameter (glycemic and lipidemic parameters) be these groups (obese vs nonobese).
- Figure legends (e.g., for Figures 1 and 2) are somewhat brief and do not adequately highlight key findings. It would be beneficial to clarify whether there are significant differences between groups in these figures.
Res. For both Fig. 1 & 2, all the details are mentioned in the legend, including the statistical tests, and explanation for the components of the figures, the p-values are presented within the figures very clearly, however, we added to the legend of figure 2, a statement that all the p-values are > 0.05 and corrected the name of the statistical test, Mann-Whitney Rank Sum Test
- Some references (e.g., References 21 and 22) show discrepancies with the study’s conclusions.
Res. Explanations are added. In the former study [21], the sample size was too small (a total of 75 lean and obese subjects), and there was no consideration for age which we believe is a major determinant for inflammation, while the latter study [22] was conducted in mice’
- The manuscript mentions that obesity may induce localized inflammation rather than systemic inflammation, but the discussion does not delve deeper into this possible mechanism.
Res. Because we didn’t test the local inflammation, it is mentioned just as one possible explanation, however, we added a paragraph about the role of local inflammation on systemic inflammation to the introduction.
- Further discuss the potential local inflammation in adipose tissue and its possible effects on systemic inflammation.
Res. As mentioned above, that is added to the introduction
- Some scientific terms, like “IL-6 up-regulates glucose uptake in insulin-dependent tissues,” are vague and could be more specific to avoid ambiguity.
Res. IL6 enhances glucose uptake in insulin-sensitive tissues, e.g., muscle, adipose tissue. That is revised
- The authors should discuss in greater detail the lack of elevation in IL-6 in obese subjects and explore potential reasons behind it. This may involve discussing the limitations of the current study, such as the methods of measurement or the specific characteristics of the sample group.
Res. This is added to the discussion and the limitations of the study but in brief, because we have no explanation, we presented it as an observation
- The findings contribute to the understanding of obesity-induced inflammation. However, there are areas for improvement, particularly in sample size, the rationale for biomarker selection, data presentation, and discussion depth.
Res. Agree, we add these points to the end of the discussion as future plans
Round 2
Reviewer 3 Report
Comments and Suggestions for Authors
This article has been greatly improved and provided excellent responses to questions.